# IMPLICIT COMPETITIVE REGULARIZATION IN GANS

## ABSTRACT

Generative adversarial networks (GANs) are capable of producing high quality samples, but they suffer from numerous issues such as instability and mode collapse during training. To combat this, we propose to model the generator and discriminator as agents acting under local information, uncertainty, and awareness of their opponent. By doing so we achieve stable convergence, even when the underlying game has no Nash equilibria. We call this mechanism *implicit competitive regularization* (ICR) and show that it is present in the recently proposed *competitive gradient descent* (CGD). When comparing CGD to Adam using a variety of loss functions and regularizers on CIFAR10, CGD shows a much more consistent performance, which we attribute to ICR. In our experiments, we achieve the highest inception score when using the WGAN loss (without gradient penalty or weight clipping) together with CGD. This can be interpreted as minimizing a form of integral probability metric based on ICR.

## 1 INTRODUCTION

**Generative adversarial networks (GANs):** (Goodfellow et al., 2014) are a class of generative models based on a competitive game between a *generator* that tries to generate realistic new data, and a *discriminator*, that tries to distinguish generated from real data. In the original formulation, the two players are playing a zero-sum game with the loss function of the generator given by the binary cross entropy,

$$\min_{\mathcal{G}} \max_{\mathcal{D}} \frac{1}{2}\mathbb{E}_{x \sim P_{\text{data}}}\left[\log \mathcal{D}(x)\right] + \frac{1}{2}\mathbb{E}_{x \sim \mathcal{G}}\left[\log\left(1 - \mathcal{D}(x)\right)\right]. \tag{1}$$

Here, $\mathcal{G}$ is the probability distribution generated by the generator, $\mathcal{D}$ is the classifier provided by the discriminator, and $P_{\text{data}}$ is the target measure, for example the empirical distribution of the training data. In practice, both players are parameterized by neural networks that are trained simultaneously by a variant of stochastic gradient descent (SGD).

**Instability and mode collapse:** GANs constitute the state of the art (SOTA) on many problems, but suffer from a variety of issues such as instability and *mode collapse*, i.e. a drastic loss of sample diversity, during training.
Beginning with (Goodfellow et al., 2014), GANs have often been interpreted as the generator performing approximate minimization of a function defined implicitly by maximization over the discriminator. From this perspective, instability and mode collapse have been explained with the fact that for finite amounts of data and any continuous distribution $\mathcal{G}$ produced by the generator, a sufficiently powerful discriminator will be able to fully maximize the two summands in Equation 1 independently of each other, instead of searching for a trade-off. This is because it can first maximize the second summand and then modify its value in $\epsilon$-neighborhoods of the true data to maximize the first summand. In the limit of small $\epsilon$, these modifications don't influence the second objective, leading to the discriminator maximizing both objectives. While one could hope that this problem vanishes for large data sets, real data will often be concentrated around low-dimensional structure, which can be exploited by the discriminator in a similar fashion. Thus, for any fixed generator we expect the weights of the discriminator to diverge as it fully performs the above trick, causing instability and preventing the generator from receiving useful gradient information.

**Gradient penalties need to choose a metric on sample space:** The above observation has motivated restriction of the size of the gradients of $\mathcal{D}$ (as a function on sample space) (Arjovsky & Bottou, 2017; Arjovsky et al., 2017; Gulrajani et al., 2017; Roth et al., 2017; Kodali et al., 2017;

Miyato et al., 2018). Measuring the size of the gradient of $\mathcal{D}$ requires a metric on the space of samples, typically chosen to be $\ell_2$. For complex samples like images, this will be a crude measure of similarity. In images of human faces for instance, skin– and hair color will likely dominate over other subtle features like facial expression in terms of the $\ell_2$-norm. In contrast, the original GAN has the intriguing property of not requiring any explicit choice of distance on sample space.

**The game behind GANs:** For convex-concave games, Nash equilbria provide a natural solution concept. We merely need to choose which algorithm to use when computing them. For non-convex-concave games like GANs however, Nash equilibria need not be the right solution concept (see Berard et al. (2019) and Figure 2), which spurred a search for new solution concepts (Jin et al., 2019; Fiez et al., 2019).

Both the choice of a loss function and of a notion of solution is a modeling problem. We argue that rather than deriving an algorithm based on these choices, it might be preferable to directly model iterative games between the two players. Just like "the goal is to capture the other player's king" does not fully specify the rules of chess, Equation 1 does not fully specify a game between the two players. Rather, we need to specify what information the players have access to, which "moves" are allowed, and in what order are they are to be made. Even with the rules completely specified, the behavior of the players will crucially depend on the heuristics that they use to anticipate each other's behavior and choose their own strategy, possibly in the face of uncertainty. Finally, a chess player might prefer a certain draw over an uncertain victory depending on the format of the tournament. Similarly, players aiming to minimize an averaged loss might play differently from players that greedily minimize the loss after every move.

Even for the same loss function, different such assumptions lead to different dynamics in GANs. Rather than trying to define a suitable solution concept based on, for instance, the spectrum of the Hessian, it might be more fruitful to directly model the behavior of the two players.

**Our contributions:** In this work we show that modeling the generator and discriminator as agents making decisions under uncertainty and in awareness of each others' goals can lead to stable behavior. Surprisingly, this holds even in the absence of Nash equilibria and for games that are meaningless in the minimax sense. We show empirically that even generators that produce good samples can incur large losses under a discriminator optimized for this particular generator, but such discriminators are highly vulnerable to changes of the generator's strategy (see Figure 2). Therefore, a discriminator aware of the competitive nature of the game will try to find a trade-off between achieving a low loss and being robust to the actions of the generator, and vice versa. Based on this observation, we propose to mitigate the lack of stability in GAN training by choosing more realistic models of the agents' behavior. One realization of this idea is the recently proposed competitive gradient descent (CGD) algorithm (Schäfer & Anandkumar, 2019), where players update their parameters using the Nash equilibrium of local approximations of the loss functions that takes into account actions of the other player. We show that it imposes an *implicit competitive regularization* (ICR) on the updates, leading to moves that are locally robust to the other player's response.

We conclude with an extensive suite of experiments using Adam and CGD on the original GAN loss from Equation 1 and the bilinear WGAN loss (Arjovsky & Bottou, 2017; Arjovsky et al., 2017), and a variety of different regularizers. We observe that the performance of CGD is much more consistent than that of Adam and that the highest inception score is achieved by combining the WGAN loss (without gradient penalty or weight clipping) with CGD. Recall that when combined with a Lipschitz constraint, WGAN minimizes the Wasserstein distance between generator and target distribution. By analogy, the combination of WGAN loss with CGD can be interpreted as minimizing an integral probability metric (Sriperumbudur et al., 2009) based on ICR. Thus, its superior performance supports the claim that ICR is the appropriate form of regularization for GANs. We emphasize that in our experiments we did not perform any architecture or hyperparameter tuning, and instead use a model intended to be used with WGAN gradient penalty (Gulrajani et al., 2017).

## 2 THE NASHLESS GAME

**What is the solution, if there is no solution?** We believe that rather than viewing the loss function of a GAN as specifying a well-defined computational problem that is solved by computing a local Nash equilibrium using one of the many methods proposed in the literature, the choice of the algorithm should part of the modeling problem. In order to make this more concrete, we consider the

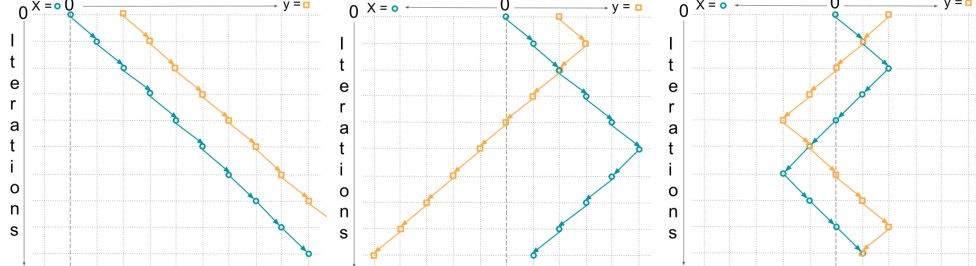

Figure 1: **Implicit competitive regularization in the nashless game:** Under full information, each player moves towards infinity as quickly as possible (first panel). Under limited information, but without accounting each other's actions, the players oscillate and eventually diverge (second panel). Under both limited information and awareness of the opponent, the trajectories become stable (third panel). (The vertical axis measures "time", each horizontal grid line corresponds to one iteration).

zero-sum game given by

$$\min_{x\in\mathbb{R}}\max_{y\in\mathbb{R}} -\exp(x^2) - \alpha xy + \exp(y^2), \qquad (2)$$

which we will refer to as *the nashless game*. Here, $\alpha \gg 1$ is a large but fixed parameter. This game does not have a global Nash equilibrium, since each player can always achieve an exponentially increasing reward by moving towards infinity. It furthermore does not have any local Nash equilibria, since the curvature of the objective function is always negative in $x$ and positive in $y$. One approach would be to interpret it as a Stackelberg game where player $y$ gets to make their move after player $x$. However, since player $y$ can always achieve arbitrarily large rewards by diverging to infinity this does still not result in a notion of solution. (Jin et al., 2019, Definition 14) introduced a notion of *local minmax point* $(x^*, y^*)$ that amounts to restricting the movement of the follower to within a small distance of $y^*$. However they also require the objective to be a local minimum with respect to $y$, which prevents it from being adequate for the game. (Fiez et al., 2019, Definition 4) propose a notion of differential Stackelberg equilibrium. However, since it is only based on first order information it will implicitly use the *worst* strategy of the follower. Thus, the leader can always locally improve by moving further towards infinity, preventing the game from having a differential Stackelberg equilibrium in the sense of (Fiez et al., 2019, Defininition 4.). While it is possible to define other notions of equilibrium, and hence solutions, for problems like in Equation 2, the above discussion illustrates that it is much harder than in the convex-concave setting.

An alternative way of turning equation 2 into an algorithm is to turn it into an iterative game between the two players, and obtain the algorithm from a suitable notion of rational behavior.

**Three games for one loss function:** To illustrate the different dynamics that can arise, we now equip the loss function of Equation 2 with three different sets of rules for iterative play.

1. *The global game*: The two players move simultaneously, a distance of at most 1. They have access to the entire optimization landscape and aim to minimize their average loss in the asymptotic limit of infinite time.

2. *The myopic game*: The two players move simultaneously at a distance of at most 1. At the $k$-th step, the player $x$ ($y$) only has access to objective function on $[x_k - 1, x_k + 1] \times \{y\}$ ($\{x\} \times [y_k - 1, y_k + 1]$) and tries to minimize the loss at step $k + 1$.

3. *The predictive game*: The two players move simultaneously at a distance of at most 1. At the $k$-th step, both players have access to the objective function on $[x_k - 1, x_k + 1] \times [y_k - 1, y_k + 1]$) and try to minimize their loss at step $k + 1$. The players are furthermore aware of each other's goals.

In the global game, player $y$ can win the game with a loss of $-\infty$ by diverging to infinity as quickly as possible. This is similar in spirit to the maximization over $\mathcal{D}$ in the original GAN where the global optimization over $\mathcal{D}$ results in unstable behavior of the underlying game (see Figure1)

Since the global optimization of nonconvex problems is computationally infeasible, the first set of rules is not a suitable model for GAN training. In myopic game, we model the use of local algorithms by only giving the players access to the loss function in a local neighborhood of their present position. Initially of the exponentially increasing reward at infinity they try to improve their

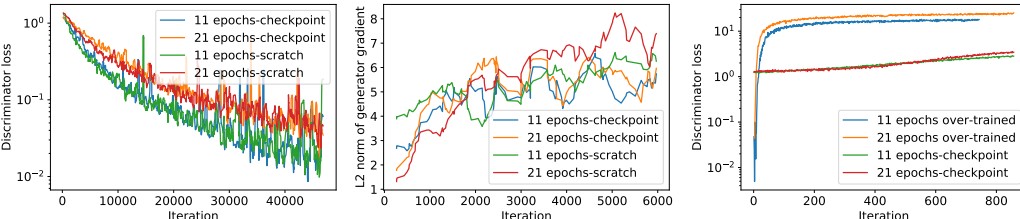

Figure 2: **Overtraining Discriminators:** We begin by training a GAN on MNIST using Adam for 21 (11) epochs, saving the resulting generator and discriminator as a "checkpoint". We then *over-train* the discriminator while keeping the checkpoint generator fixed, achieving low discriminator loss (second panel). As indicated by the increasing gradient norm of the generator during training (third panel), the resulting discriminator is brittle and incurs large losses as we start training the generator again (fourth panel). Values computed from minibatches and averaged over small $x$-windows.

reward locally, entering the cycling dynamics also known from simultaneous gradient descent (see Figure 1). While the cycling behavior slows down the divergence, the players will eventually diverge far enough for the exponential term to overpower the bilinear term, from where on both players will diverge to infinity, without further cycling behavior.

In predictive game, the two players have access not only to their own, but also to the other player's loss function as well as the effect that the other player's moves will have on their own loss. Therefore, they don't blindly aim to increase their own loss function but also try to estimate the actions of the other player and adapt choose their own actions accordingly. In the initial position $x = 0$, $y = 2$, a player $y$ that is unaware of the presence of $x$ will choose to move to 3, in order to minimize its loss. If $y$ is aware of the goals of $x$ however, it will know that whatever move $y$ makes, the best move for $x$ will be to move to 1. Thus, player $y$ should make its move based on the assumption that $x$ will move to 1 and hence (since $\alpha \gg 1$) moves to 1 instead. For the given starting conditions there always exist unique optimal strategies for $x$ and $y$ resulting in the periodic behavior illustrated in Figure 1.

The above example shows that even in the absence of suitable equilibria we can obtain meaningful and stable dynamics by modeling the players as agents acting with limited information and awareness of their opponents. Before providing a possible framework for this type of modeling, we will provide evidence that key features of the nashless game might indeed be present in GANs.

## 3   WHY A COMPETITION-AWARE GAN MIGHT BE STABLE

**GAN discriminators are not optimal:** The key feature of the nashless of the last section is that even though either player is incentivized to diverge to infinity, in doing so it enables the other player to "counter-attack" by means of the strong bilinear term. In order to verify these properties in GANs, we first train a GAN on MNIST using 21 iterations of Adam, obtaining a good generative model. For future reference, we will refer to this generator and discriminator as the *checkpoint* generator and discriminator. We then *overtrain* a new discriminator from scratch against the *fixed* checkpoint generator. Similar to Arjovsky & Bottou (2017), we observe that the overtrained discriminator can achieve orders of magnitude lower discriminator loss than the checkpoint discriminator. Therefore, the checkpoint discriminator is far from being globally optimal. It seems conceivable that the checkpoint discriminator might instead be locally optimal. In order to verify this claim, we continue training the checkpoint discriminator while keeping the checkpoint generator fixed. As shown in Figure 2, we observe almost the same behavior as when overtraining the discriminator from scratch. Thus, the checkpoint discriminator was not at a local minimum (that is deep enough to affect stochastic gradient descent). Recent results by Berard et al. (2019) indicate that many good (in terms of image quality) generators are not local minima, but rather critical points of their loss function. This casts further doubt on the idea that local Nash equilibria are the right notion of solution for GANs.

**Optimal discriminators in GANs are brittle:** Given that the discriminator could so drastically improve over the checkpoint generator we have to ask ourselves why it doesn't do so during GAN training. What, if anything, makes the checkpoint discriminator the better choice over the over-

trained one? In the nashless game, players were partly discouraged from decreasing their losses since in doing so they became would become more vulnerable to counter play by the other player. To investigate this question we now perform a second experiment were we fix the overtrained discriminators from the first experiment and start training the checkpoint generator again. As shown in Figure 2, the discriminator loss increases by two orders of magnitude within just 40 iterations, significantly beyond the loss at the checkpoint. In contrast, training the generator against the fixed checkpoint discriminator the discriminator loss does not show a significant increase over the first 100 iterations. This behavior is already foreshadowed by the increase in norm of the generator's gradient in the initial stage of the overtraining procedure (see Figure 2). Thus, while the overtrained discriminator can achieve a greatly reduced loss, it is much more vulnerable to changes of the generator. A discriminator that is aware of the goals of the generator might therefore choose not to minimize its loss myopically but rather aim for a trade-off between achieving a low loss and being robust to actions of the generator.

## 4 MODELING THE PLAYERS' ACTIONS

**Belief, uncertainty, and anticipation:** In the previous section we observed that GANs, just like the nashless game, might not be described well by the solution concept of a (local) Nash equilibrium. However, our experiments suggested that a suitable notion of rational play of the two players might indeed lead to stable GAN formulations. As described in the introduction and further illustrated on the nashless game, there are numerous ways to turn a given loss function for the two players into an iterative game. In the following, we will present one possible framework towards this goal. We will restrict our framework towards the case of both players moving *simultaneously*, *forgetting* the history of the game and *greedily* aiming to minimize their loss after each move. Our framework models the players as choosing optimal actions based on the following three ingredients.

1. The *belief* that the agents have about the loss associated with different actions.
2. The way they incorporate their *uncertainty* into their decisions.
3. Their *anticipation* of the action of the adversary when making a decision.

In the myopic version of the nashless game, both players' model consisted of the knowledge of the loss function in a 1-neighborhood of their own position. The agents handled their uncertainty conservatively, by refusing to step outside the region where they are aware of the loss function. The agents did not have enough information to make use of an anticipation of the other agent's strategy, thus we can take them to assume that the other player stays fixed. The next iterate $(x_{k+1}, y_{k+1})$ can therefore be obtained as solution to the optimization problem

$$
\min_x f(x, y_k) + \phi_{\{x:|x-x_k|\leq 1\}}(x)
$$
$$
\max_y f(x_k, y) - \phi_{\{y:|y-y_k|\leq 1\}}(y),
$$

(3)

where $f$ is the loss function and $\phi_A(x) = \infty$ for $x \notin A$ and zero everywhere else.

In the predictive version of the nashless game, both players' model consists of the true loss function in a 1-neighborhood of the present position of either player. As before, the players treat the uncertainty conservatively by refusing to step outside the region where they have knowledge of the objective function. Differently from the myopic game, they are aware of the objectives of the other player and the way in which the other player's actions affect their own loss. Therefore, their update is obtained as the Nash equilibrium of

$$
\min_x f(x, y) + \phi_{\{x:|x-x_k|\leq 1\}}(x)
$$
$$
\max_y f(x, y) - \phi_{\{y:|y-y_k|\leq 1\}}(y).
$$

(4)

In the particular case considered in the nashless game the above local game always had a pure equilibrium. For general problems, the optimal solution of the local problem might correspond to a randomized strategy, resulting in a randomized algorithm for the "solution" of the two-player game.

**Computationally feasible instances:** In practice, we usually don't have access to the exact loss function in a local neighborhood, but rather to a local approximation that we expect to less reliable

as we move away from the present iterate. A natural and computationally feasible way of modelling agent behavior in this setting is to let the agents use a Taylor approximation of the loss function, together with a regularizer that penalizes moving too far away from the last iterate, and thus into regions where the Taylor approximation is unreliable. The popular simultaneous gradient descent falls into this framework, with players playing optimal with respect to a linear approximation of the loss and a quadratic penalty modeling their uncertainty, obtaining the game

$$\min_x f(x_k, y_k) + (x - x_k)^\top \nabla_x f(x_k, y_k) + \frac{1}{2\eta} \|x - x_k\|^2$$

$$\max_y f(x_k, y_k) + (y - y_k)^\top \nabla_y f(x_k, y_k) - \frac{1}{2\eta} \|y - y_k\|^2.$$

By choosing different models for the player's uncertainty, we could obtain other instances of simultaneous mirror descent. This game is very similar to the one in equation 3 and simultaneous gradient descent can be seen as a computationally feasible version of the myopic game and as such features a similar oscillatory behavior.

In order to mitigate this problem, we can allow the players to anticipate each others actions, for instance by assuming that the other player will perform a gradient descent update. For the players to be able to use this information we have to introduce a bilinear approximation, obtaining *Learning with opponent-learning awareness* (LOLA) (Foerster et al., 2018) as solution to the local game

$$\min_x f(x_k, y_k) + (x - x_k)^\top \nabla_x f(x_k, y_k) + (x - x_k)^\top D^2_{xy} f(x_k, y_k) \eta \nabla_y f(x_k, y_k) + \frac{1}{2\eta} \|x - x_k\|^2$$

$$\max_y f(x_k, y_k) + (y - y_k)^\top \nabla_y f(x_k, y_k) - (y - y_k)^\top D^2_{xy} f(x_k, y_k) \eta \nabla_x f(x_k, y_k) - \frac{1}{2\eta} \|y - y_k\|^2.$$

Optimistic mirror descent (Daskalakis et al., 2017) and the extragradient method (Korpelevich, 1977) can similarly be interpreted as based on "optimistic" predictions of the two players actions.

**Competitive gradient descent:** In LOLA, both players make their decision based on the assumption that the other player will play gradient descent. In contrast, the predictive variant of the nashless game lets the players choose the Nash equilibrium of the local problem, thus accounting the uncertainty on the other player's plans. A computationally feasible analogue of this idea is given by *competitive gradient descent* (CGD) (Schäfer & Anandkumar, 2019), which lets the players choose the Nash equilibrium of a bilinear approximation of the loss function, regularized with a quadratic penalty. The resulting game is given by

$$\min_x \ f + (x - x_k)^\top \nabla_x f + (x - x_k)^\top D^2_{xy} f (y - y_k) + (y - y_k)^\top \nabla_y f + \frac{1}{2\eta} \|x - x_k\|^2$$

$$\max_y \ f + (x - x_k)^\top \nabla_x f + (x - x_k)^\top D^2_{xy} f (y - y_k) + (y - y_k)^\top \nabla_y f - \frac{1}{2\eta} \|y - y_k\|^2,$$

where all derivatives and function values of $f$ in the above expression are evaluated in the last iterate, $(x_k, y_k)$. Schäfer & Anandkumar (2019) observe that this game has a pure unique Nash equilibrium resulting in the update rule

$$x_{k+1} = x_k - \eta \left( \mathrm{Id} + \eta^2 D^2_{xy} f D^2_{yx} f \right)^{-1} \left( \nabla_x f + \eta D^2_{xy} f \nabla_y f \right)$$

$$y_{k+1} = y_k + \eta \left( \mathrm{Id} + \eta^2 D^2_{yx} f D^2_{xy} f \right)^{-1} \left( \nabla_y f - \eta D^2_{yx} f \nabla_x f \right).$$

**Implicit competitive regularization in CGD:** While Schäfer & Anandkumar (2019) introduced CGD as a method for finding Nash equilibria, we believe that by modeling the behavior of the two players it also provides a useful notion of implicit regularization. Indeed, while the gradient term $\nabla_x f$ models the goal of $x$ to minimize $f$. The additional term $\eta D^2_{xy} f \nabla_y f$ is a gradient step for the penalty term $\eta \|\nabla_y f\|^2$, similar to LOLA or symplectic gradient adjustment (SGA) (Gemp & Mahadevan, 2018; Balduzzi et al., 2018; Letcher et al., 2019) which can be interpreted as attempting to decrease the effect of the other player's action. The term $\left( \mathrm{Id} + \eta^2 D^2_{xy} f D^2_{yx} f \right)^{-1}$ corresponds to choosing updates that are robust to the actions of the other player. If some of the singular values of $D_{xy} f$ are very large, this amounts to approximately restricting the update to the orthogonal complement of the corresponding singular vectors. For smoothly varying singular vectors, this can be though of as approximately constraining the trajectories to a manifold of robust

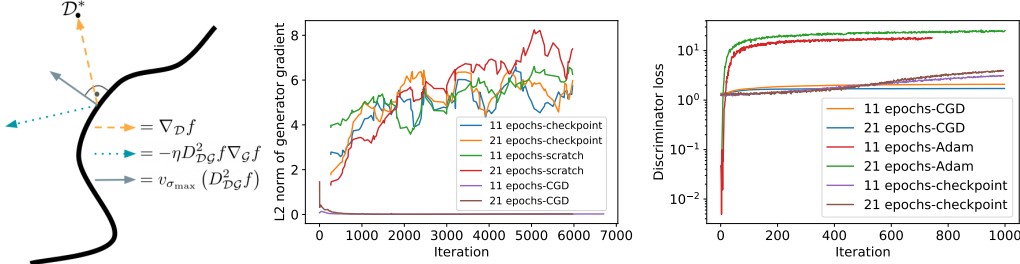

Figure 3: **Implicit competitive regularization:** First panel: While simple gradient ascent would lead towards the overtrained discriminator $\mathcal{D}^*$ (first arrow), CGD tries to also reduce the other players gradient (second arrow) and for large mixed Hessian $D^2_{\mathcal{D}\mathcal{G}}f$, projects the update on the orthogonal complement of the leading singular vector. We illustrate how this could prevent overtraining, with the thick black curve depicting the "manifold of robust play". Second and third panel: When attempting to overtrain the discriminator using CGD, it actually becomes *more* robust (compare to Figure 2).

play (see Figure 3 for an illustration). Contrary to the explicit regularization using gradient penalties, this *implicit competitive regularization* does not require choosing a metric on the space of samples but is instead informed by the inductive biases of the generator's and discriminator's architecture. We will now investigate how the use of CGD affects the overtraining in on MNIST presented in Figure 2. To this end, we try to overtrain the discriminator from the checkpoint, this time using CGD instead of Adam. As shown in Figure 3, no significant overtraining takes place even after 21 epochs. In fact, the generator gradients are *smaller* and the discriminator is *more robust* than at the checkpoint.

## 5 EMPIRICAL STUDY ON CIFAR10

**Experimental setup:** One of the key claims in this work is that the implicit competitive regularization due to CGD is enough to allow stable GAN training. To substantiate this claim, we compare Adam and CGD on a wide range of regularizers and loss functions. While it is still unclear what is the best way of combining CGD with learning rate heuristics, we opt for a naive combination of CGD with RMSprop (referred to as ACGD) to make for a more fair comparison to Adam. As loss functions, we use the original GAN loss (OGAN) of equation 1 and the Wasserstein GAN loss function (WGAN) given by

$$\min_{\mathcal{G}} \max_{\mathcal{D}} \ \mathbb{E}_{x \sim P_{\text{data}}} \left[ \mathcal{D}(x) \right] - \mathbb{E}_{x \sim P_{\mathcal{G}}} \left[ \mathcal{D}(x) \right].$$

When using Adam on OGAN, we stick to the common practice of replacing the generator loss by $\mathbb{E}_{x \sim \mathcal{G}} \left[ -\log \left( \mathcal{D}(x) \right) \right]$, as this has been found to improve training stability (Goodfellow et al., 2014; 2016). In order to be generous to existing methods, we use an existing architecture intended for the use with WGAN gradient penalty (Gulrajani et al., 2017). As regularizers, we consider no regularization (NOREG), $\ell_2$ penalty on the discriminator with different weights (L2), Dropout ($p = 0.5$) on the discriminator after each convolutional layer (Dropout), or 1-centered gradient penalty on the discriminator, following Gulrajani et al. (2017) (GP). Following the advice in (Goodfellow et al., 2016) we train generator and discriminator simultaneously, with the exception of WGAN-GP and Adam, for which we follow (Gulrajani et al., 2017) in making five discriminator updates per generator update. We use the Pytorch implementation of inception score (Salimans et al., 2016) to quantitatively compare the quality of the different generators.[1]

**Experimental results:** We will now summarize our main experimental findings. **(1:)** When restricting our attention to the top performing models, we observe that the combination of ACGD with the WGAN loss and without any regularization achieves higher inception score than all other combinations tested (see Figure 4). **(2:)** Many of the other high-performing models are obtained via ACGD

---

[1]Note that a Pytorch implementation results in slightly different scores compared to a Tensorflow implementation. We also report Tensorflow inception scores for important runs in Appendix 7.1 that show that the relative performance of the different models remains largely the same.

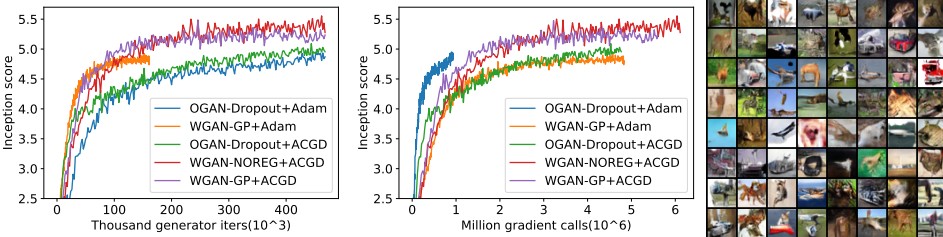

Figure 4: We plot the inception score against the number of iterations (first panel) and gradient or Hessian-vector product computation (second panel). The third panel shows final samples of WGAN trained with ACGD and without regularization.

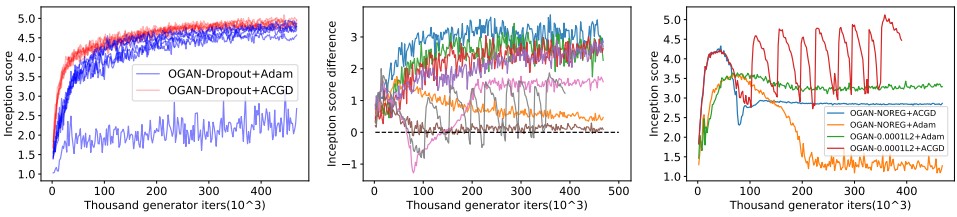

Figure 5: First panel: We compare the variance over seven runs of the same model with ACGD and Adam. Second panel: We plot the difference between inception scores between ACGD and Adam (positive values correspond to a larger score for ACGD) over all iterations and models. Third panel: The only cases where we observe nonconvergence of ACGD are OGAN without regularization or with weight decay of weight 0.0001. The inception score is however still higher than for the same model trained with Adam.

(see Figure 4). **(3:)** When comparing the number of gradient computations and Hessian-vector products, ACGD is significantly slower than OGAN combined with ADAM and dropout, because of the iterative solution of the matrix inverse in ACGD's update rule (see Figure 4). **(4:)** The only instance where we observe erratic behavior with ACGD is when using OGAN without regularization, or with a small $\ell_2$ penalty. However, ACGD still outperforms Adam on those cases (see Figure 5). We note that we observed nan values in one case of OGAN-L2 and ACGD, presumably because of underflow in the CG solve as discriminator weights approach zero. **(5:)** When plotting seven runs of OGAN with dropout using both ACGD and Adam, the results of ACGD are consistent while Adam shows a large variance (see Figure 5). **(6:)** When plotting the difference between the inception score obtained by ACGD and Adam for the same model over the number of iterations, for all models, we observe that ACGD often performs significantly better, and hardly ever significantly worse (see Figure 5).

## 6 CONCLUSION AND OUTLOOK

In this work we have argued that difficulties in GAN training might better be addressed by modeling the behavior of generator and discriminator than by just modifying the loss function or devising new local solution concepts, like generalizations of Nash or Stackelberg equilibria. We have proposed a framework for doing so and illustrated how it relates to different algorithms in the literature. For one instance of our framework, competitive gradient descent, we explore in more detail the implicit competitive regularization that it induces. Extensive numerical experiments show that the performance of competitive gradient descent is significantly more consistent than that of Adam across different loss functions and regularizers. In particular, the highest performing model is the bilinear WGAN loss together with competitive gradient descent *and no explicit regularization*. Analogous to the Wasserstein distance being the *integral probability metric* (IPM) (Sriperumbudur et al., 2009) associated to a Lipschitz constraint, the WGAN loss optimized with CGD can be intuitively interpreted as a form of IPM associated to implicit competitive regularization.
GANs allow us to employ neural networks for generative modeling without having to prescribe a notion of similarity between images explicitly, by taking full advantages of their "inductive biases". The present works suggests that their training might be best stabilized by sticking to this approach

and relying on implicit competitive regularization rather than using penalties that make an explicit choice of distance function on sample space. We believe that once reliable methods for their training have been established, adversarial methods will greatly increase the scope of applications of deep neural networks.

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

# 7 APPENDIX

## 7.1 TENSORFLOW INCEPTION SCORE

We computed Tensorflow inception scores for important runs of our experiments. As reported in the figure 6, our results match the ones reported in the literature (Figure 3 in Gulrajani et al. (2017)) with ACGD still outperforming WGAN-GP trained with Adam.

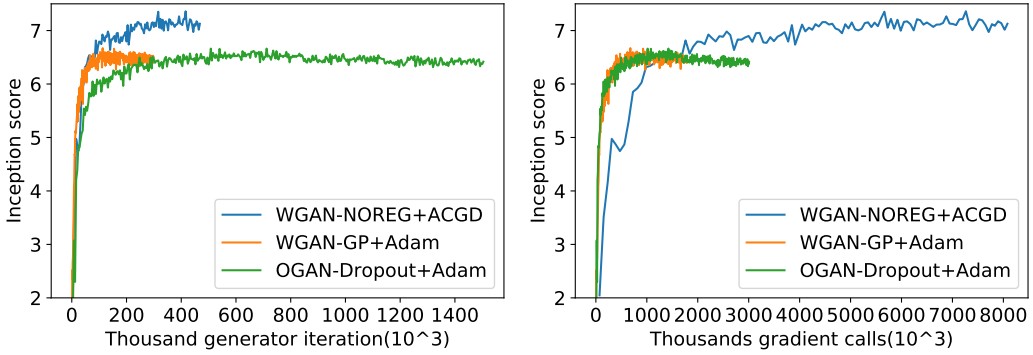

Figure 6: We plot the Tensorflow inception scores against the number of generator iterations(first panel) and gradient or Hessian-vector product computation (second panel).

