# OpenReview forum: "Implicit competitive regularization in GANs"
_ICLR.cc/2020/Conference — Reject_

### Official Review · AnonReviewer1 · 2019-10-23
**Official Blind Review #1**

**Rating:** 6

**Review:**

The paper presents a new way of regularization in Generative Adversarial Network (GAN). It is well known that a naive training of GAN can fail to converge. Although GAN is relatively a new concept, many papers tried to introduce a good way of stabilizing GAN training. I believe that this paper is addressing the stability issues in the most fundamental and effective ways. The paper utilizes Competitive Gradient Descent proposed by Schäfer and Anandkumar in 2019 in training GAN. The intuition is that both players should predict what their opponent is going to do. This results in a convergence point where each agent becomes robust against changes of the other agent. The performance of the new method was demonstrated on CIFAR10.

The paper is definitely interesting. If this method works as well as the authors claim, it can significantly improve the practicality of GAN. The paper is very readable and understandable but many small typos and grammar errors can be found in the text. This can be easily corrected by the authors.

However, the contribution of this paper is questionable. The original CGD paper already applies it to train a GAN.

I would also appreciate if the method is tested on multiple other data sets.

Overall, the paper is well-written, technically correct and interesting enough for the venue. However, as I pointed out above, the contribution should be more clearly stated.



**Experience Assessment:**

I have read many papers in this area.

**Review Assessment: Checking Correctness Of Derivations And Theory:**

I assessed the sensibility of the derivations and theory.

**Review Assessment: Checking Correctness Of Experiments:**

I assessed the sensibility of the experiments.

**Review Assessment: Thoroughness In Paper Reading:**

I read the paper at least twice and used my best judgement in assessing the paper.

---

> ### Author Response · Authors · 2019-11-13
> **Response to review #1**
>
> We thank the reviewer for their positive assessment and helpful feedback.
>
> We believe that the contribution of the present work is complementary to that of the initial work on CGD. While CGD proposes an algorithm for general competitive optimization problems, the present work investigates how the modeling of generator and discriminator as agents acting under uncertainty and opponent awareness can stabilize GANs, even for seemingly unstable loss functions.
> CGD is a particularly clean instance of this phaenomenon, which is why we study its implicit competitive regularization in more detail.
> However, the insights into the stabilization of GAN training, which we consider the main contribution of the present work, are not restricted to CGD. Instead, we hope that they encourage a new approach to devising optimization algorithms for GANs. See also our response to Reviewer #2.

---

### Official Review · AnonReviewer2 · 2019-10-24
**Official Blind Review #2**

**Rating:** 6

**Review:**

The paper presents a novel training methodology for GANs to improve stability. The resulting regularization, termed implicit competitive regularization, updates the parameters of both the generator and the discriminator to be robust to one another. A framework for practical application of this approach is described -- this is done by a local Taylor approximation of the loss and updating each model’s parameters to this approximate model’s nash equilibrium.  The method is shown to prevent overfitting and produce high-performing models with consistent training.

The approach and insights are reasonable and the problem is worthwhile to approach. The method is clear and the associated code is appreciated. The results are interesting in terms of describing the ICR property and demonstrating its performance.

The paper gives background and intuition to solidifying the CGD update. Are there additional algorithmic approaches that are possible and potentially more efficient with this understanding in mind? This I believe would help solidify the paper and build beyond CGD.

Some additional results that could clarify the benefits:
- A primary contribution of the training approach is training consistency. The distribution over many training runs should be provided in figures.
- Clearly due to the additional gradient calls the approach is computationally slower, as shown in Figure 4. If each approach is trained for the same amount of time, how does the performance compare?
- One may expect that the update may result in more conservative updates and thus potentially lower-performing policies in the limit. If there iterations were instead log-scale to show performance in high training iterations, is there any loss of performance in top-performing runs?
- Could a similar approach be used to allow safe gradient updates according to a risk over the opponent’s possible updates, e.g., via CVar? This may also be a stable training procedure as well with less conservatism.

The paper should be proofread, there are several minor typos throughout, e.g.:
- “generators producing that produce good” -> generators that produce good
- “This game is very similar similar” -> repeated word
- “GAN trainin.”


**Experience Assessment:**

I do not know much about this area.

**Review Assessment: Checking Correctness Of Derivations And Theory:**

I assessed the sensibility of the derivations and theory.

**Review Assessment: Checking Correctness Of Experiments:**

I assessed the sensibility of the experiments.

**Review Assessment: Thoroughness In Paper Reading:**

I read the paper at least twice and used my best judgement in assessing the paper.

---

> ### Author Response · Authors · 2019-11-13
> **Response to review #2**
>
> We thank the reviewer for their positive assessment and helpful feedback.
>
> Regarding
> ==============================================================
> ”The paper gives background and intuition to solidifying the CGD update. Are there additional algorithmic approaches that are possible and potentially more efficient with this understanding in mind? This I believe would help solidify the paper and build beyond CGD.”
> ==============================================================
>
> Indeed, CGD is just one instance of implicit regularization that maps particularly well onto the discussions in section 2 and 3,
> Our hope is that this work spurs the development of new algorithms that emphasize the game-theoretic modeling of generator and discriminator acting under uncertainty and awareness of their opponent.
> The framework we give in section 4 is meant to serve as a starting point to this end, and also to  relate our work to existing methods in the literature, such as LOLA. Since a discussion of all the different algorithms that could be built on these ideas is beyond the scope of the paper, we restrain ourselves to treating CGD, as an example.
>
>
> Regarding the remaining comments:
> - Repeating the entire experiments over many runs is very computationally expensive, which is why we singled out one setting (OGAN-DROPOUT) to repeat over seven runs (first panel of figure 5). If there are other configurations that you would find particularly enlightening to see over multiple runs, we will be happy to add them to the paper.
>
> - Under optimal implementation (mixed mode automatic differentiation), the number of backward passes provided in the second panel of figure 4 should provide a good proxy for the time complexity. As we can see in this plot, for different time budgets, different methods will yield better results. In the limit of large time budgets however, ACGD-WGAN seemed to perform best.
> Our experiments in figures 4 and 5 suggest that in the limit of many training iterations, models trained with ACGD tend to saturate at higher inception score.
>
> - This is a nice idea and a great example of the kind of thinking we wanted to encourage with this work!

---

### Official Review · AnonReviewer3 · 2019-11-01
**Official Blind Review #3**

**Rating:** 8

**Review:**

The paper analyzes instability in training GANs, relates it to Nash equilibria, and proposes a novel training set-up based on competitive nashless games. The solution is related to other recently proposed work, but the paper brings additional insights into understanding it.

The analysis on conditions that lead to divergence or convergence, and of the proposed solution, are interesting. I recommend accepting.

I have some basic knowledge of GANs but am not deeply familiar with the field. The paper was accessible to me on a high level. Especially compelling to me were sections 2 and 3. The empirical study also seemed to yield positive results.

Suggestions for improvement:
- Additional proofreading would be beneficial.
- The scale of the axes in figure 1 is not clear, making it a little less compelling.
- Inception score is used as the only evaluation metric for the generators. Perhaps this is standard in the field, although human ratings would seem more reliable to me.

**Experience Assessment:**

I do not know much about this area.

**Review Assessment: Checking Correctness Of Derivations And Theory:**

I did not assess the derivations or theory.

**Review Assessment: Checking Correctness Of Experiments:**

I assessed the sensibility of the experiments.

**Review Assessment: Thoroughness In Paper Reading:**

I made a quick assessment of this paper.

---

> ### Author Response · Authors · 2019-11-13
> **Response to review #3**
>
> We thank the reviewer for the positive assessment. In particular, we are very pleased to hear that the reviewer found Sections 2 and 3 compelling, since we consider them central to our work.
> We also thank the reviewer for the suggestions for improvement.
> We are doing additional proofreading and added a sentence describing the vertical axis in Figure 1. While we agree that human scores would be ideal this is unfortunately not practical, which is why we settled for the inception score, which is a popular measure of sample quality in the field.

---

### Official Review · AnonReviewer5 · 2019-11-02
**Official Blind Review #5**

**Rating:** 1

**Review:**

https://openreview.net/pdf?id=SkxaueHFPB

The paper has some interesting ideas but I don’t think any of them are fully fleshed out.

I find the reporting of Inception Score highly suspect. The authors choose WGAN-GP as a baseline and report scores of ~4.5 vs ~5.5 with their modification. However the WGAN-GP paper reports an IS of 7.86 on CIFAR. Furthermore, current GAN SOTA on CIFAR is approaching IS=9. I am not making the argument that the authors ought to demonstrate SOTA results, however they should at least present results which are consistent with the published results of their chosen baseline.

The authors then make this statement:
“Thus, its superior performance supports the claim that ICR is the appropriate form of regularization for GANs. We emphasize that in our experiments we did not perform any architecture or hyperparameter tuning, and instead use a model intended to be used with WGAN gradient penalty”
This does not hold, since the numbers reported are far below the actual baseline.

Besides this major point, I am unconvinced by some of the mathematical statements in the paper. Much of the mathematical details are deferred to the original CGD paper. It is not really particularly reader-friendly to defer that to the CGD paper since they are seemingly crucial to the discussion here. Relative to the CGD paper some signs have been flipped and some definitions appear to be used in subtly different ways which makes for a very difficult read. I feel that far too much has been left as an exercise to the reader.

Concretely my concerns refer to the main discussion of the effect of the CGD as a regularizer:

The authors state:
“If some of the singular values of Dxy are very large, this amounts to approximately restricting the update to the orthogonal complement of the corresponding singular vectors”

I don’t see how this is the case. The terms Dxy/Dyx aren’t really introduced or defined anywhere in this work. Assuming is the transpose of the other (?) then the update direction is:
A + B where A=inv(S) grad_x and B = inv(S)Dxy grad_y (and S = I + Dxy Dyx). So we have a term which is being affected by the smallest singular values of S and a term which is the orthogonal projection of grad_y onto Dxy, alternatively the ridge-regression fit of grad_y on Dxy which would attenuate directions corresponding to the *small* singular values (as is well known from the theory of ridge regularizers). I feel like there is much more to say here than what is discussed in the paper in very vague terms.

Of course the effective rank of S, or the rate of decay of its singular values is crucially important. In practise I would assume the smaller SVs of Dxy to be difficult to estimate or the matrix to be rank deficient in which case they would simply be unity in the inverse whereas the directions corresponding to large singular values would be attenuated. So in this case it is the regularized orthogonal complement but its not clear (if the matrix is not full rank) that it is a meaningful direction (and again this is all highly dependent on the effective rank, too).

Further on it is mentioned: “For smoothly varying singular vectors, this can be though of as approximately constraining the trajectories to a manifold of robust play”.

First it is not at all clear to me what “smoothly varying singular vectors” are. Varying with respect to what? Secondly, the “manifold of robust play” has not been defined anywhere.

Finally, figure 3 is quite bizarre to me. None of the quantities have been rigorously defined and so it seems like the relative effect of each of the arrows and the manifold have been drawn arbitrarily in order to fit the story, rather than to actually illuminate the true behaviour in an intuitive manner.
B (defined above) has a very clear interpretation as a least-squares fit so I figure that any geometric interpretation of the CGD update direction could start from there.

**Experience Assessment:**

I have published one or two papers in this area.

**Review Assessment: Checking Correctness Of Derivations And Theory:**

I assessed the sensibility of the derivations and theory.

**Review Assessment: Checking Correctness Of Experiments:**

I assessed the sensibility of the experiments.

**Review Assessment: Thoroughness In Paper Reading:**

I read the paper at least twice and used my best judgement in assessing the paper.

---

> ### Author Response · Authors · 2019-11-06
> **Experiments on inception scores**
>
> Thank you for your review. We disagree with your assessment but believe that it can be explained by misunderstandings as outlined below. We will add additional clarifying remarks along the detailed response below that will hopefully prevent this from happening in the future.
>
> We will first address the concerns about the experiments on inception scores:
>
> The review reads:
> ===========================================================
> “I find the reporting of Inception Score highly suspect. The authors choose WGAN-GP as a baseline and report scores of ~4.5 vs ~5.5 with their modification. However the WGAN-GP paper reports an IS of 7.86 on CIFAR. Furthermore, current GAN SOTA on CIFAR is approaching IS=9. I am not making the argument that the authors ought to demonstrate SOTA results, however they should at least present results which are consistent with the published results of their chosen baseline.”
> ===========================================================
>
> The inception score of 7.86 in the WGAN-GP paper was achieved with a larger ResNet architecture and measured using the tensorflow inception score (IS) (Table 3 in the WGAN-GP paper).
> For our experiment, we used an existing pytorch port ( https://github.com/EmilienDupont/wgan-gp/blob/master/models.py ) of the DCGAN structure in the WGAN-GP repository ( https://github.com/igul222/improved_wgan_training/blob/master/gan_cifar.py ) that was used to produce Figure 3 in the WGAN-GP paper and which is only reported to achieve IS of about 6, and reported the pytorch IS. The pytorch IS can be ~5-10% lower than the tensorflow IS (this is  reported, for instance, by https://github.com/ajbrock/BigGAN-PyTorch/blob/master/inception_utils.py ), which puts us in the ballpark of the results reported in the WGAN-GP paper. We are in the process of computing tensorflow IS and FIDs for select runs.
>
> The review furthermore reads:
> ===========================================================
> “The authors then make this statement:
> “Thus, its superior performance supports the claim that ICR is the appropriate form of regularization for GANs. We emphasize that in our experiments we did not perform any architecture or hyperparameter tuning, and instead use a model intended to be used with WGAN gradient penalty”
> This does not hold, since the numbers reported are far below the actual baseline.”
> ===========================================================
>
> As outlined above the numbers that we report for the baseline methods are similar to those in the literature. We furthermore emphasize that the scientific purpose of these experiments is to test whether implicit competitive regularization (ICR) is present in GANs and whether it provides a more appropriate way of regularizing GAN training. To this end we show WGAN-loss without regularization is not only stable when using CGD (which runs against the arguments proposed in the original WGAN papers), but that CGD also leads to improved performance and robustness compared to other optimizers and regularization methods.
> For this argument, it is more important that the architecture and parameters are reasonable and not cherrypicked (which is why we choose them from the literature on WGAN-GP), rather than whether they achieve “SOTA” inception score.

---

> ### Author Response · Authors · 2019-11-06
> **Discussion of why CGD acts as a regularizer**
>
> Please see below for our response to the concerns regarding the discussion why CGD acts as a regularizer:
>
> The review states:
> ===========================================================
> “The authors state:
> “If some of the singular values of Dxy are very large, this amounts to approximately restricting the update to the orthogonal complement of the corresponding singular vectors”
>
> I don’t see how this is the case. The terms Dxy/Dyx aren’t really introduced or defined anywhere in this work. Assuming is the transpose of the other (?) then the update direction is:
> A + B where A=inv(S) grad_x and B = inv(S)Dxy grad_y (and S = I + Dxy Dyx). So we have a term which is being affected by the smallest singular values of S and a term which is the orthogonal projection of grad_y onto Dxy, alternatively the ridge-regression fit of grad_y on Dxy which would attenuate directions corresponding to the *small* singular values (as is well known from the theory of ridge regularizers). I feel like there is much more to say here than what is discussed in the paper in very vague terms.”
> ===========================================================
>
> We apologize for omitting the definition of the $D_{xy}^2f$, which refers to the matrix containing the partial derivatives $\frac{\partial^2f}{\partial x_{i} \partial y_{j}}$ of the objective function. Under mild regularity assumptions, $D_{xy}^2$ and $D_{yx}^2$ are indeed transposes of each other (Schwarz’s theorem).
> Both A and B attenuate directions corresponding to **large** singular values of $D_{xy}^2$. The intuitive explanation is that they both have a square of $D_{xy}^2$ appearing in the denominator (i.e as a matrix inverse), but at most a single factor of $D_{xy}$ appearing in the numerator.
> By developing $x$ and $y$ in the basis given by the left resp. right singular vectors of $D_{xy}^2$ this argument is made rigorous by observing that the component of $\nabla_x$ that corresponds to the singular value $\sigma$ is attenuated by a factor $1/(1+\sigma^2)$ while the corresponding component of $\nabla_y$ is attenuated by a factor $\sigma/(1 + \sigma^2)$.
> We note that $D_{yx}  B$ and not B is the (regularized) projection of grad_y onto the row space of Dxy.
>
> The review further states:
> ===========================================================
> “Of course the effective rank of S, or the rate of decay of its singular values is crucially important. In practise I would assume the smaller SVs of Dxy to be difficult to estimate or the matrix to be rank deficient in which case they would simply be unity in the inverse whereas the directions corresponding to large singular values would be attenuated. So in this case it is the regularized orthogonal complement but its not clear (if the matrix is not full rank) that it is a meaningful direction (and again this is all highly dependent on the effective rank, too).
>
> Further on it is mentioned: “For smoothly varying singular vectors, this can be thought of as approximately constraining the trajectories to a manifold of robust play”.
>
> First it is not at all clear to me what “smoothly varying singular vectors” are. Varying with respect to what? Secondly, the “manifold of robust play” has not been defined anywhere.”
> ===========================================================
>
> The projection onto the orthogonal complement of $D_{xy}f$ is meaningful since it corresponds to strategies of one player, the effect of which (to leading order) does not depend on the move of the other player.
> In an attempt to provide additional intuition, we propose to think of these directions as constituting the tangent space of a “manifold of robust play”, that is a manifold of strategies on which the players can move around by only playing strategies that are robust in the sense that their payoff is (to leading order) unaffected by arbitrary simultaneous moves of the other player.
> We apologize if this intuition was not helpful and will try to add additional clarifications.
>
> Lastly, the review states:
> ===========================================================
> “Finally, figure 3 is quite bizarre to me. None of the quantities have been rigorously defined and so it seems like the relative effect of each of the arrows and the manifold have been drawn arbitrarily in order to fit the story, rather than to actually illuminate the true behaviour in an intuitive manner.
> B (defined above) has a very clear interpretation as a least-squares fit so I figure that any geometric interpretation of the CGD update direction could start from there.”
> ===========================================================
>
> The first panel of figure 3 is indeed not a plot, but an illustration of how the two regularization terms in CGD could prevent overtraining, proposing an explanation for the empirical observation that CGD does not “overtrain” the way that Adam does. We apologize that this was not clear and will add some additional clarification.

---

> ### Author Response · Authors · 2019-11-10
> **Update: Tensorflow inception score**
>
> We have rerun ACGD-WGAN-NOREG and ADAM-WGAN-GP and computed the official tensorflow inception scores, as opposed to the pytorch one. ADAM-WGAN-GP now reaches inception score of approximately 6.5, consistent with the scores reported in Figure 3 of the WGAN-GP paper. (See our reply below for a more detailed discussion).
>
> ACGD-WGAN-NOREG reaches tensorflow inception score of approximately 7.1-7.3 and thus still improves upon ADAM-WGAN-GP, just as with the Pytorch inception score. We expect the remaining conclusions of our experiments to stay the same when changing from pytorch to tensorflow inception score, although rerunning all experiments may take some time.
> Plots, code, and model can be accessed under https://drive.google.com/drive/folders/10M0keSN47PfWi4-L6bdQfXWx4BSTs2c3

---

### Author Response · Authors · 2019-11-13
**Response to all reviewers**

We would like to thank all reviewers for their assessment and feedback. The main purpose of this work was to illustrate a novel mechanism that could stabilize GAN training without the need for Lipschitz-regularization (for instance, through gradient penalties) and we were happy to see that this idea was appreciated by most reviewers.
It seemed to us that there were two main concerns from the side of the reviewers:

(1:) Reviewer #5 pointed out that the inception score for WGAN-GP in our publication is significantly below the one reported in the original WGAN-GP publication. As detailed below this was due to the use of a pytorch-- as opposed to tensorflow implementation of inception using a slightly different model. We computed tensorflow inception scores for  WGAN-GP and CGD and observe that they match the results reported in the literature with CGD still outperforming WGAN-GP trained with Adam (for more details, see our response to reviewer #5).

(2:) Reviewers #1 and #2 were concerned about the extent to which our contribution goes beyond the existing work on CGD. We emphasize that the main contribution of this work is the presentation of a novel mechanism for the stabilization of GAN training, of which CGD is just one manifestation. In particular, most of section 4 describes a general framework for the utilization of implicit competitive regularization. By modelling the information accessible to the agents, their way of handling uncertainty, and their anticipation of the other players action, multiple existing algorithms can be recovered. We furthermore hope that this framework will be useful in guiding the development of novel algorithms.

We are thankful for any further comments.

---

### Decision · Program_Chairs · 2019-12-19

**Decision:**

Reject

**Comment:**

The paper proposes to study "implicit competitive regularization", a phenomenon borne of taking a more nuanced game theoretic perspective on GAN training, wherein the two competing networks are "model[ed] ... as agents acting with limited information and in awareness of their opponent". The meaning of this is developed through a series of examples using simpler games and didactic experiments on actual GANs. An adversary-aware variant employing a Taylor approximation to the loss.

Reviewer assessment amounted to 3 relatively light reviews, two of which reported little background in the area, and one more in-depth review, which happened to also be the most critical. R1, R2, R3 all felt the contribution was interesting and valuable. R1 felt the contribution of the paper may be on the light side given the original competitive gradient descent paper, on which this manuscript leans heavily, included GAN training (the authors disagreed); they also felt the paper would be stronger with additional datasets in the empirical evaluation (this was not addressed). R2 felt the work suffered for lack of evidence of consistency via repeated experiments, which the authors explained was due to the resource-intensity of the experiments.

R5 raised that Inception scores for both the method and being noticeably worse than those reported in the literature, a concern that was resolved in an update and seemed to center on the software implementation of the metric. R5 had several technical concerns, but was generally unhappy with the presentation and finishedness of the manuscript, in particular the degree to which details are deferred to the CGD paper. (The authors maintain that CGD is but one instantiation of a more general framework, but given that the empirical section of the paper relies on this instantiation I would concur that it is under-treated.)

Minor updates were made to the paper, but R5 remains unconvinced (other reviewers did not revisit their reviews at all). In particular: experiments seem promising but not final (repeatability is a concern), the single paragraph "intuitive explanation" and cartoon offered in Figure 3 were viewed as insufficiently rigorous. A great deal of the paper is spent on simple cases, but not much is said about ICR specifically in those cases.

This appears to have the makings of an important contribution, but I concur with R5 that it is not quite ready for mass consumption. As is, the narrative is locally consistent but quite difficult to follow section after section. It should also be noted that ICLR as a venue has a community that is not as steeped in the game theory literature as the authors clearly are, and the assumed technical background is quite substantial here. For a game theory novice, it is difficult to tell which turns of phrase refer to concepts from game theory and which may be more informally introduced herein. I believe the paper requires redrafting for greater clarity with a more rigorous theoretical and/or empirical characterization of ICR, perhaps involving small scale experiments which clearly demonstrates the effect. I also believe the authors have done themselves a disservice by not availing themselves of 10 pages rather than 8.

I recommend rejection at this time, but hope that the authors view this feedback as valuable and continue to improve their manuscript, as I (and the reviewers) believe this line of work has the potential to be quite impactful.